# Transboundary Collaborative Modeling: Consensual Identification and Ranking of Flood Adaptation Measures—A Case Study in the Mono River Basin, Benin, and Togo

Adrian Delos Santos Almoradie [1,*], Nina Rholan Houngue [1] , Kossi Komi [2], Julien Adounkpe [3] and Mariele Evers [1]

[1] Department of Geography, University of Bonn, 53115 Bonn, Germany; rholan.houngue@uni-bonn.de (N.R.H.); mariele.evers@uni-bonn.de (M.E.)
[2] Laboratory of Research on Spaces, Exchanges and Human Security, Department of Geography, University of Lomé, Lomé 01BP1515, Togo; kossik81@yahoo.fr
[3] Laboratory of Applied Ecology, Faculty of Agronomic Sciences, University of Abomey-Calavi, Abomey-Calavi P.O. Box 526, Benin; julvictoire@yahoo.com
* Correspondence: adrian.almoradie@uni-bonn.de

**Abstract:** The field of environmental management, specifically flood risk management (FRM), emphasizes participatory decision-making to address diverse issues and conflicting interests among stakeholders. This approach recognizes the complexity of decisions and their long-term impact on sustainability. Collaborative knowledge production is crucial for understanding the system, generating scenarios, and establishing consensus on mitigation and adaptation measures. Transboundary FRM requires an interdisciplinary and transdisciplinary approach, employing suitable tools and methods for assessment and decision-making. In the context of the CLIMAFRI project, funded by the German Federal Ministry of Education and Research (BMBF), we evaluated the practicality of a participatory Collaborative Modeling framework in the transboundary Lower Mono River (LMR) basin, shared by Togo and Benin. This framework enables holistic understanding, stakeholder engagement, and the identification of appropriate adaptation-mitigation measures based on predefined evaluation criteria and a Multi-Criteria Decision Method. Our study customized and evaluated the framework considering the impact of the COVID-19 pandemic, which restricted face-to-face interactions. The study's results indicate that in both countries, FRM is characterized as being more proactive rather than preventive, meaning the actions taken mainly address a broader range of potential issues and opportunities rather than targeting specific risks to minimize their impact. Moreover, it is crucial to enhance preventive measures and further improve the flood assessment capacity. The information obtained from scenarios involving the Adjarala Dam, land-use, and climate change under RCP 4.5 and 8.5 scenarios for the years 2030, 2050, and 2100 is valuable for decision-making regarding the development and prioritization of adaptation measures. The ranking of the seven measures shows that capacity building is the most preferred, followed by dykes, early warning systems, regulation of land use, insurance, and retention zones. The group ranking of Togo and Benin highlights differences in their perceptions and interests, where Togo leans towards soft measures, while Benin prefers hard (physical) measures.

**Keywords:** collaborative modeling; Mono River Basin; adaptation measures; transboundary flood risk management

## 1. Introduction

The register of international river basins based on the studies done by McCraken and Wolf (2019) [1] using new data and changes in political boundaries has identified 310 transboundary basins shared by 150 countries (https://transboundarywaters.science.oregonstate.edu/content/register-international-river-basins, accessed on 25 May 2023). Covering 47.1% of the Earth's total surface area, these transboundary basins accommodate

52% of the world's population. With half of the Earth's population living in transboundary basins, it is evident that transboundary river floods are a major issue, especially in mitigating their impact across countries. A global study from 1985–2005 on the documentation of flood events shows that transboundary floods affected larger areas and were severe in magnitude, and this accounted for 32% of casualties and 60% of affected people [2]. This exemplifies that the impact of transboundary flooding is considerable and that this may exacerbate in the future because of population growth, urbanization, and the change of climate that will increase people's exposure and vulnerability to floods. Bakker (2009) [2] also found out that transboundary floods in developed countries, compared to developing countries, have the greatest financial damage but have the fewest casualties due to the difference in socio-economic factors. Among the continents, African basins bear the greatest proportion of flooding occurrences (33%), followed by South America (29%), Asia (29%), North America (21%), and Europe (19%) [3]. Moreover, six out of the 12 river basins with the highest score on flood severity are African basins, and four are South American. Many regions in Africa, especially West and East Africa, are highly vulnerable and fragile, and risk must therefore be addressed with a transboundary approach [3].

Transboundary floods consider no administrative boundaries, and their management on a basin scale is so complex that few transboundary flood risk management (FRM) mechanisms have been implemented. The need for a paradigm shift in transboundary river management, from supply management to adaptive management and from sovereign unilateralism to multilateralism, to develop measures on a basin scale is of utmost importance [4]. This shift has occurred in the European Union following the EU flood directive [5], which required member states with shared rivers to plan and develop FRM strategies at a transboundary basin level. A good example of transboundary FRM cooperation in Europe is the Rhine Action Plan on floods and the Scheldt estuary [4,6–8].

There is also some progress on transboundary FRM studies outside the EU. Many African countries have already started to accelerate transboundary cooperation under the 2015 Africa Adaptation Initiative, aiming to accelerate cooperation toward identifying adaptation actions and enhancing the understanding of risk across the continent [9]. Elsewhere, case-specific studies, such as the Kabul River Basin (Pakistan and Afghanistan) [10] and the Chenab River Basin (India and Pakistan) [11], are some examples of transboundary FRM studies in Asia. Unlike the EU approach, both of the case studies were based mainly on the modeling and investigation of future scenarios with limited or no engagement of stakeholders to understand the flooding problem, validate the models, and identify mitigation measures. This is also true for many parts of the world with transboundary river basins. This complexity arises from the challenge of convening stakeholders from different countries and bringing them together in a format to discuss FRM at a transboundary level because of varying self-interests that can oftentimes be conflicting [7]. Decision-makers encounter pressures as a result of conflicting water usage demands, and the existing methods and policies fail to adapt to the present circumstances [4].

Further challenges in transboundary FRM are in communicating, raising awareness and understanding of risk, and developing adaptation measures that are difficult to integrate and resource intensive [3,8]. To tackle this issue, one possible approach is to enhance the institutional capacities of countries and establish incentives for collaboration. This would facilitate the development of an adaptation framework to examine the viability and efficacy of adaptation measures or strategies. Moreover, employing suitable tools and methods would be crucial in effectively implementing transboundary flood risk management (FRM) and ensuring its success [4,8]. Considering the future, it is more pressing for countries with transboundary rivers to increase collaboration and reach a collective agreement due to climate change. Such collaboration will reduce the pressures of climate change on vulnerability whilst increasing resilience [7]. Adaptation measures can be better identified if the climate risks are understood and when decision-makers choose "low regret" rather than "optimal" solutions and adaptation measures that are flexible for a range of possible future scenarios.

The reviewed literature illustrates the multifaceted nature of transboundary FRM from the carrying-out cross-country engagement of stakeholders, raising awareness, communicating and understanding hazards, vulnerability, and risk, as well as the identification and selection of the right adaptation measures. The necessity for an interdisciplinary and transdisciplinary approach, along with the utilization of appropriate tools and methods, becomes evident when considering transboundary flood risk management (FRM). This comprehensive approach enables a more holistic assessment and decision-making process.

In transboundary flood risk management (FRM), the implementation of participatory, collaborative decision-making mechanisms is limited due to the complexity of convening stakeholders from different countries. This process requires significant resources, and the integration of stakeholders' perspectives becomes challenging due to conflicting self-interests, such as competing water usage priorities.

This paper introduces a study that investigates the suitability of the participatory Collaborative Modeling framework [12] in comprehensively understanding the hazard and identifying sets of adaptation measures in the transboundary Mono River Basin, which is shared by Togo and Benin. The study actively involves stakeholders in the process. The paper is organized as follows: Section 2 outlines various approaches for participatory decision-making, Section 3 demonstrates the methodologies employed in the Collaborative Modeling framework, while Section 4 deliberates on the findings. Finally, Section 5 presents the conclusion of the study.

## 2. Collaborative Modeling for Participatory Decision-Making

Encouraging participatory approaches in decision-making is a prominent aspect of environmental management, including FRM. This emphasis stems from the recognition that relying exclusively on expert knowledge is insufficient to fully grasp the complexities of intricate issues, information, and contextual factors, especially within local settings. By involving various stakeholders, informed decisions can be made by considering a broader range of perspectives and insights [13,14]. Therefore, there is a requirement for the co-production of knowledge in decision-making processes [15–17]. This involves various stages, including understanding the issues and problems, developing scenarios, and reaching a consensus on the identification and selection of mitigating measures. Recognizing the need to shift to participatory environmental decision-making, world institutions and organizations developed frameworks, guidelines, and directives such as the Aarhus Convention [18], the European Water Framework Directive (WFD) [19], and Flood Directive (FD) [5], the Flood Risk Regulations of 2009 and the Flood and Water Management Act of 2010 in the United Kingdom [20] and the World Bank's integrated urban FRM guidelines [21].

Participation in FRM can emanate in many forms, from information sharing and consultation to collaborative decision-making [22–24]. Although there is a shift to participatory FRM in many countries, it has fundamentally persisted in the form of consultative decision-making [25]. This, however, is changing to more inclusive and holistic participation in the form of participatory modeling or collaborative modeling. The term participatory modeling and collaborative modeling has been widely used interchangeably in many studies, both of which emphasize the value of engaging stakeholders in a modeling process [26]. Basco-Carrera et al. (2017) [27] attempted to make distinctions between both. Collaborative modeling can be defined as a specific type of participatory modeling that involves a higher level of collaboration, discussion, co-designing, and co-decision-making. It is particularly suitable for situations that require extensive collaboration and active involvement of stakeholders, whereas participatory modeling level of engagement is more on information sharing to consultation with some form of discussion.

Drawing the line between participatory modeling and collaborative modeling, here are some notable and recent works on FRM. For participatory modeling, citizen science approaches were used to enhance flood modeling [28,29], and flood hazard maps were

co-produced with communities and stakeholders [30–32]. On the other hand, numerous collaborative modeling approaches attempted to develop a decision support tool and set up a model by including the local knowledge and perceptions of the stakeholders [32,33]. Moreover, the engagements of stakeholders in the development of FRM and water resources management intervention options are also explored in some studies [12,34,35].

In the field of FRM, Maskrey et al. (2021) [25] stated that there are no right or wrong techniques for carrying-out collaborative modeling; it all depends on which contexts the techniques are appropriate to use to deliver in practice. Collaborative modeling can be in the form of undertaking workshops and/or focus group discussions along with decision tools to address the issue. Such examples of decision tools applied in different fields, as investigated by Maskrey et al. (2021) [25], are the use of Fuzzy Cognitive mapping in water resources management [36], System Dynamics to reduce levels of vulnerability and exposure [37], and Bayesian Network to investigate the sensitivity of measures [34]. The mentioned techniques have their strengths and limitations, and this must be recognized by considering its characteristic of being holistic, adaptable, accessible, evaluative, and transparent. Furthermore, it is ideal to integrate the full spectrum of collaboration from understanding the issue and problem scenario development to the identification and consensual decision-making on the feasible sets of measures for implementation. Although decision support tools have been encouraged and some developed for use in disaster risk management, many vulnerable countries have no or adequate decision tools. An example is Ghana's disaster risk management, many activities and initiatives have/are being done to address the issues, but the lack of decision tools makes it difficult to decide on suitable actions [38].

The Collaborative Modeling framework by Evers et al. (2012) [12] took a step further to comprehensively engage stakeholders in co-designing and co-producing knowledge through social learning. This framework is a process-based approach supported by technical tools such as models, information technology, and methods that integrate both scientific facts and stakeholders' knowledge in understanding the FRM issues in order to build consensus for the selection of mitigation measures. Due to the complexity of transboundary FRM in the LMR, the Collaborative Modeling framework cited is seen as an ideal comprehensive approach to address all FRM levels, from having a common understanding of the flooding issues and different interests to consensual decision-making on mitigating measures. The Collaborative Modeling framework, initially developed, applied, and tested by Evers et al. (2012) [12], was first implemented in the Cranbrook catchment in the United Kingdom, which faced pluvial flooding issues, and the Alster catchment in Germany, which dealt with fluvial flooding problems. The results generated a common understanding amongst stakeholders regarding flood risk, and sets of FRM measures were jointly identified and ranked for further consideration in the planning and management. The study concluded that this framework is capable of actively involving stakeholders through social learning using socio-technical instruments supported by web-based tools and methodologies, e.g., multi-criteria decision analysis [12].

## 3. Methods

### 3.1. Case Study

The Mono River Basin, situated in West Africa, is jointly shared by the Republics of Benin and Togo. Spanning across an area of 23,736.64 km$^2$, it is positioned between longitudes 0.62° E and 1.99° E and latitudes 6.28° N and 9.39° N. The basin encompasses two distinct climatic zones. The southern part experiences a sub-equatorial climate with two rainy seasons and two dry seasons, while the tropical zone has one dry season and one rainy season. Over the past three decades, the basin has recorded an average annual precipitation of 1200 mm, and the average temperature ranges from 26 °C to 28 °C. The lower part of the river experiences an annual average flow rate of 125 m$^3$/s, with a documented peak of 950 m$^3$/s. Part of the Mono River serves as a natural border between Benin and Togo (Figure 1). The basin is important for both countries in different ways: it covers

about 35% of Togo's territory; different economic activities such as agriculture, tourism, and fishing take place in the valley and along the Mono River; and the catchment hosts the Nangbéto hydroelectric dam, that is commonly owned by the two countries. The Nangbéto Dam is one of the water cooperation symbols between Benin and Togo, and it is jointly managed by the two countries through a common agency: the Communauté Electrique du Bénin (CEB). Moreover, the catchment disposes of a basin authority, the Mono Basin Authority, that was approved in 2014 and its executive direction established in 2019. Additionally, recurrent floods occur in the LMR basin, which has a flat terrain, which is defined in this study as the section of the basin situated downstream of the Nangbéto Dam. These flood events are caused by intense rainfalls and water release from the Nangbéto Dam [39]. One of the most devastating flood events in the catchment was recorded in 2010, with a maximum discharge of 950 m$^3$/s in station Athieme (downstream), causing about USD 300 million of loss and damages in the two countries [40,41]. During such extreme events, critical infrastructure, housing, and agricultural belongings (including food stored in shelters and unharvested products in the field) are the most exposed elements [42]. In addition, poverty, the proximity of farms and settlements to the river, and the lack of early warning systems were identified as the main vulnerability drivers in the LMR basin [42,43]. Moreover, recent studies based on climate and land use change scenarios in the catchment reported a potential intensification of rainfall and more frequent extreme flood events by 2050 [44,45]. In that regard, the two countries are planning a second dam, the Adjarala Dam, which is intended, among other goals, to reduce extreme flood impacts downstream. Agriculture, fishing, livestock breeding, and small-scale trades are the main economic activities in the catchment. The primary land use classification in this study consists predominantly of savanna, with croplands, forests, bodies of water, and settlements following in descending order.

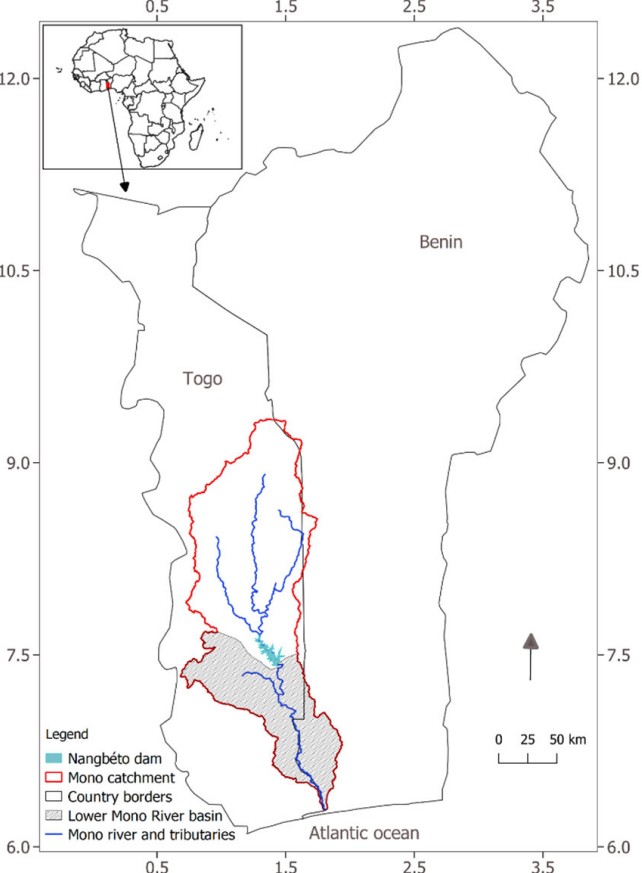

**Figure 1.** Location of the Mono River Basin.

*3.2. Transboundary Collaborative Modeling LMR Framework*

The study adapted and tested the Collaborative Modeling framework proposed by Evers et al. (2012) [12] in the transboundary LMR FRM context, taking into account the specific case settings and the impact of the COVID-19 pandemic, which limited face-to-face interactions. The modified Collaborative Modeling framework was implemented through four workshops (Figure 2) and involved five key steps. These steps included: Step 0—Defining the system: This aimed to gain a better understanding of the flooding situation and existing policies. Step 1—Current situation analysis: The focus was on comprehending the challenges faced by the current FRM practices. Step 2—Identification of flood hotspots and scenarios: This involved pinpointing the most exposed areas prone to flooding and exploring various scenarios. Step 3—Stakeholder validation: The model results were validated by the stakeholders to facilitate the identification of adaptation measures and criteria for evaluating their effectiveness. Step 4—Collaborative adaptation measure ranking: This step entailed conducting an exercise to rank the adaptation measures collaboratively, supporting consensus-based decision-making regarding preferences for implementation. All workshops were conducted in the official French language, which is spoken in both Togo and Benin.

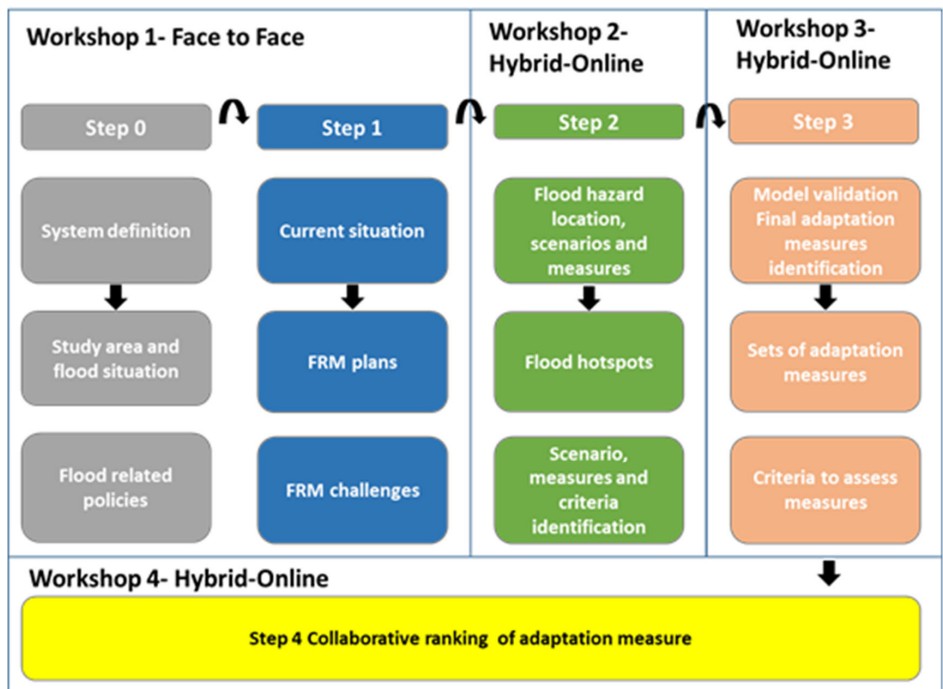

**Figure 2.** LMR Collaborative Modeling Framework-Adapted from Evers et al. (2012) [12].

The study identified key important stakeholders through our African partner networks and expert interview and surveys. The main criteria for identifying the stakeholders are those with key roles in the policy, planning, management, research, decision-making, and most active engagement in flood disaster risk response. Furthermore, in the implementation of the Collaborative Modeling, diverse stakeholders were engaged to holistically include different knowledge and interest (sometimes diverging) in the decision-making process, categorizing them as emergency responders, planning and implementation, research, and Non-Government-Organization.

Supporting the discussions and engagement of stakeholders, numerous tools and products were used to communicate flood hazards, scenarios of return periods as well as climate and land use change [44,46]. For spatio-temporal products on flood hazard base cases and scenarios, the Telemac-2D and Soil Water and Assessment Tool (SWAT) were used as the hydrodynamic and hydrological models, respectively. For details about the LMR hazard study, refer to Houngue et al. (2023) [45].

Due to COVID-19 pandemic travel restrictions, Workshops 2 to 4 were carried out in a hybrid format (online and face-to-face amongst stakeholders). The ZOOM web conferencing tool was used for communication in a plenary and for conducting a break-out group discussion. The Mentimeter tool was used for an interactive question and answer that required individual feedback (with confidentiality), polling options, and visualization. For presenting the hazard maps and mapping the flood hotspots with the stakeholders interactively, Google Maps was used. This tool was selected due to the familiarity with the use of the participants and workshop organizers. To ensure a more substantial engagement in case technical issues arise, such as participants experiencing internet connectivity problems, the hybrid format was designed to take place within a half-day timeframe. The main challenge lies in actively involving the participants in an interactive manner. To maintain a high level of engagement, the strategy employed was to have a moderator present both in the physical room and online for those participating remotely.

Carrying out the ranking of measures in an exercise, the stakeholder participants were grouped into three, country-based (Benin and Togo) and diversified (Benin-Togo). The third mixed group comprises stakeholders from both countries who participated online via Zoom. Figure 3 is the workflow of the ranking of measures. The first step is for the stakeholder group to agree and to provide weights on the identified criteria (sometimes also called objectives) by considering the criteria's relative importance based on their preference and expert judgment. This is done by distributing budget points from 0 to 100, having an overall sum of 100. The second step is to evaluate quantitatively or qualitatively the adaptation measures against the criteria. Quantitative evaluation (cannot be modified) is centered on the results of the flood hazard model, and qualitative evaluation is linguistic terms (e.g., low, medium, and high) that stakeholders have to indicate based on their perception of how the adaptation measure performs. The qualitative terms were then converted to crisp numbers using the MCDM fuzzy set theory conversion scale [47]. Lastly, the measures are ranked with the Technique for Order Preference by Similarity to Ideal Solution (TOPSIS) by Hwang and Yoon (1981) [48] from most to less preferred measure. TOPSIS is a multi-criteria decision method (MCDM) interchangeably, also referred to as Multi-Attribute Decision Method (MADM), that uses the notion of distances from ideal points (or ideal solutions) to come up with the ranking. The method was used for ranking because of its simplicity, transparency in decision-making, and ease of adaptation in comparison to other methods. The ranking of adaptation measures is then discussed amongst the group to agree if it is satisfactory. If there is disagreement, iteratively, the group may rethink the given weight on the criteria and the evaluation of adaptation measures performance and run the TOPSIS calculation again. The groups then presented the results in a plenary to compare their rankings and for open discussion in a more transboundary setting.

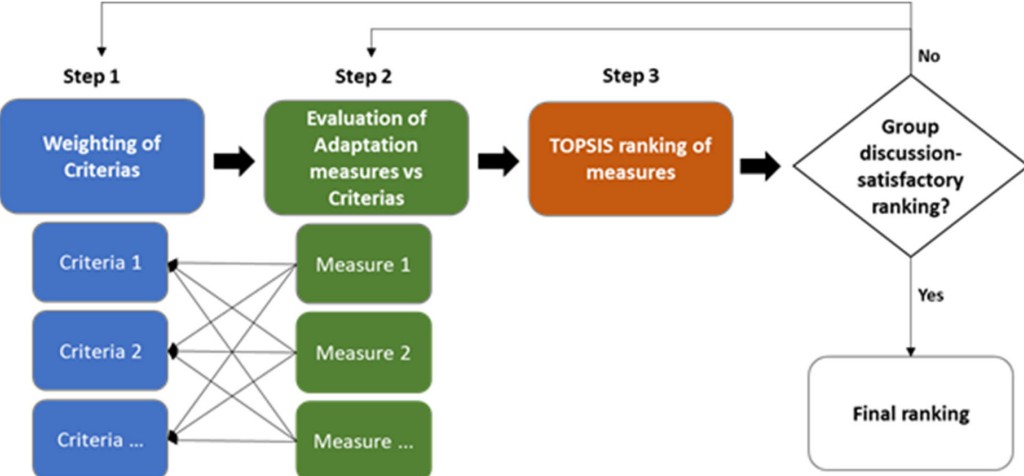

**Figure 3.** Workflow of the ranking of measures.

A customized web portal was developed for this ranking exercise to interactively evaluate and rank the adaptation measures. The web portal has four main parts for stakeholders to go through:

- Part 1—Hazard map, measures, and criteria: this is for participants to become familiar with hazard information (for reference case 2010 and representative concentration pathway RCP4.5 with return periods of 2 and 100 years), the measures and criteria identified for this exercise.
- Part 2—Decision matrix: here, the participants have to assign weights to each criterion and then assess each measure against every criterion on a qualitative basis.
- Part 3—Ranking of adaptation measures: based on the weighting and the assessment from Part 2.
- Part 4—Finally, one can check the assessment and computation data in the summary section of the portal.

### 3.3. Stakeholders

Through our African partner networks and expert interviews and surveys, the study identified 37 stakeholder institutions that have a key role in LMR FRM from both Benin and Togo. The stakeholders were identified based on their crucial roles in policy, planning, management, research, decision-making, and active involvement in responding to flood disaster risks. These institutions worked from the level of community, response, planning, and management up to the policy. In conducting the workshops, for a more cohesive and productive engagement among stakeholders, participants were kept in a smaller group (15–20 stakeholders) that would allow for an in-depth discussion, more transparency, and ensure that their views and perceptions have been considered.

Benin stakeholders:

1. Research/Academia
   West African Science Service Centre on Climate Change and Adapted Land Use (WASCAL-Benin); National Institute for Agricultural Research of Benin (INRAB); Pan African University Institute of Water and Energy Sciences (PAUWES)
2. Implementation, Policy, and Administration
   National Agency for Civil Protection (ANPC); Ministry of Agriculture; Ministry of Environment (MCVDD); Directorate for Water; Ahemey dev. lake Agency (ADELAC); Meteo-Benin; Municipality of Athieme; Mono Basin Authority Benin representative; National Fund for Climate and Environment (FNEC)
3. NGO and Insurances
   Caritas; Red Cross; Young Volunteers for the Environment (JVE-Benin); Divine Miséricorde H2; CIF Assurances: National Water Partnership (PNE); German Society for International Cooperation (GIZ)

Togo stakeholders:

1. Research/Academia
   WASCAL-Togo; Institute of Research Agronomic Research (ITRA);
2. Implementation, Policy, and Administration
   ANPC; Meteo; Ministry of Communication; Ministry of Commerce; Nangbeto Dam Authority; Ministry of Agriculture; Ministry of Infrastructure and Transport; Ministry of Environment; Mono Basin Authority Togo representative;
3. NGO and Insurance
   Eau Vive; JVE-Togo; Caritas; SUNU Assurance; SAHAM Assurances; Fidelia Assurances; Bank Mondiale.

### 3.4. Workshop Structure and Implementation

The workshops were implemented between June 2019 and March 2022. The Kickoff workshop was a face-to-face physical presence held in Lome prior to COVID-19 travel restrictions. However, for the other three workshops, there was a need to adapt to engaging

the stakeholders due to the COVID-19 pandemic situation. The three workshops were conducted in a hybrid-online format where researchers (experts/modelers) engaged the participants online and had stakeholders in-person in a room and online, both in Benin and Togo. Workshop 1 to 3 was designed and implemented to achieve a common shared understanding of the FRM in the LMR and to frame the needed specifics and facts for the ranking of adaptation measures. This ranking was realized in workshop 4. Table 1 is an overview of the workshop's aim and activities.

**Table 1.** Summary of workshop aims and activities.

| Workshop Aim | Activities |
|---|---|
| 1st Workshop—Kickoff face-to-face (June 2019) | • Present the overall objectives of the study and how it can contribute to addressing the current and future flood situation in the LMR; <br> • Share and discuss the current status and cooperation initiatives of transboundary FRM in the LMR; <br> • Information about the current flood adaptation measures and policies; <br> • Share relevant challenges in FRM; <br> • Learn from stakeholders' expertise and experience; <br> • Field visits to complement literature information. |
| 2nd Workshop Hybrid-Online (August 2020) Preliminary identification of flood hazard "hotspots", adaptation measures, scenarios, and criteria/objectives to assess the performance of measures | • Identify areas (hotspots) exposed to flooding along the LMR to support the validation of the flood hazard model developed; <br> • Identify an initial set of scenarios that could modify flood risk in the LMR catchment; <br> • Derive potential adaptation measures as input for modeling scenario-based impacts of flooding; <br> • Identify initial sets of criteria/objectives to assess the performance of adaptation measures as the basis for ranking them afterward. |
| 3rd Workshop Hybrid-Online (January 2021) Model verification, climate, and land-use scenarios definition, and preliminary identification of adaptation measures and criteria | • Present the results of climate scenarios in the Mono River Basin; <br> • Verify the results of the flood model with stakeholders; <br> • Identify with stakeholders the final set of adaptation measures and criteria for upcoming collaborative modeling. |
| 4th Workshop Hybrid-Online (March 2022) Participatory exercise on the ranking of adaptation measures for possible implementation | • Present the measures identified in the previous workshops as well as the process for selecting the measures included in the collaborative modeling exercise; <br> • Present and demonstrate, through an exercise with stakeholders, a consensual decision-making approach for the selection of potential adaptation measures to be implemented. <br> • Consider and understand perceptions on the ranking of measures from an individual and group perspective |

Workshop 1 organized in the city of Lomé, Togo, was attended by 12 stakeholders from Benin and Togo for a day of activity. Stakeholders were first engaged in a breakout group (country based) to avoid any influence from stakeholders from each other's country. Then lastly, both countries, together in a plenary, presented the outcome of the group to discuss their similar or divergent knowledge, views, interest, and understanding of FRM, initiative actions and challenges, and potential contribution of the research study in the LMR FRM. This also provided an opportunity to develop synergies on the different issues related to flooding and its management. Conceivably the first round of stakeholder identification may have overlooked some key actors, a round of discussion was also initiated to identify other institutions that needed to be involved in FRM. Workshop 2, in a hybrid-online format, was country-based and carried out in a two-day workshop (one day for each country). The workshop engaged 14 and 12 stakeholders from Benin and Togo, respectively. The objective was to first gather information on the location of the most exposed and vulnerable areas to flooding ("hotspots") using a web-based GIS for model validation. Moreover, also to gain in-depth insight into the difference or similarities in the views and interests of the stakeholders of each country regarding the identification of preliminary sets of scenarios, potential adaptation measures, and criteria to assess the performance of measures.

Workshop 3, also in a hybrid format, was organized in a one-day session having all stakeholders from both countries engaged in the same virtual room. This was participated by 15 stakeholders from Benin and 12 from Togo. This workshop aimed to synthesize and finalize the common scenarios, adaptation measures, and criteria identified from Workshop 2 to prepare the final information needed for the participatory exercise for the last workshop. Scenarios on climate and land-use change and the results of the flood model with the flood hotspots were also presented to further validate the model with the local knowledge of stakeholders. Workshop 4, similar to the last workshop, was carried out by having both countries in a virtual room. The session ranked the measures through an exercise followed by a presentation and discussion of the evaluation and rank. There were 9 stakeholder participants from Benin and 10 from Togo. For the collaborative modeling exercise, the stakeholders were organized into three groups, stakeholders attending face-to-face in Benin and Togo and the participants of both countries joining online. This approach provides us with an understanding of the views of stakeholders from both countries regarding the performance of measures through the ranking results and their evaluation. Additionally, having a group all together from both countries provides us with some insights into the difference in the ranking through consensual decision-making. The results of the different groups were then presented in the plenary virtual room for further discussion and reflection with all stakeholder participants.

## 4. Results and Discussion

### 4.1. Steps 0–3: Shared Understanding of Flood Risk, Scenarios and Its Management

The success of participatory FRM and decisions taken depends on the process of carrying-out stakeholder engagement and on the outcome of having all stakeholders jointly discuss and have a common understanding of the issues, current situation, possible scenarios, and intervention measures. In this study, workshops have been pragmatically planned, structured, and implemented to achieve the objectives of Collaborative Modeling. Moreover, process-based customized presentation of information aided by tools was seen as necessary to effectively convey the results of the models produced by the experts/modelers who are involved in the process. Here we present the outcome of Steps 0–3, which was supported by numerous technical tools.

#### 4.1.1. Steps 0 and 1-System Definition and Current Situation

Steps 0 and 1, carried out in workshop 1, show that for both countries, FRM is more proactive than preventive. Both countries anticipate flooding using an early warning system (EWS) model and community gauges. It was found that the EWS model needed improvement because of a lack of data. Response and recovery are systematized; this is, however, constrained because of the lack of flood hazard and risk maps and insufficient resources. There is also a need to increase preventive measures and to further enhance the capacity of flood assessment based on scientific facts to support decision-making. Benin and Togo's flood risk policies are made public through sensitization by means of policy briefs for civil protection and national strategy for disaster risk reduction.

#### 4.1.2. Step 2-Flood Hazard Mapping, Scenarios, Measures, and Criteria

Step 2, implemented in workshop 2, was structured into a four-part session. First, stakeholders identified the areas most exposed to flooding, followed by the identification of scenarios and initial sets of measures and criteria. The following presents the results of the sessions.

Part 1. Identification of areas exposed to flood (flood hotspots)

This session presented the preliminary results of flood modeling and demonstrated the tools (google maps and Mentimeter) for them to gain access and know how to use the online maps in the identification of flood-exposed areas. This was designed to support the building of the flood model and to validate its outcome.

The mapping exercise confirmed the preliminary output of the flood model, showing flood hotspots mainly along the LMR, specifically downstream. Togo and Benin stakeholders identified 41 specific villages or districts affected (see Figure 4).

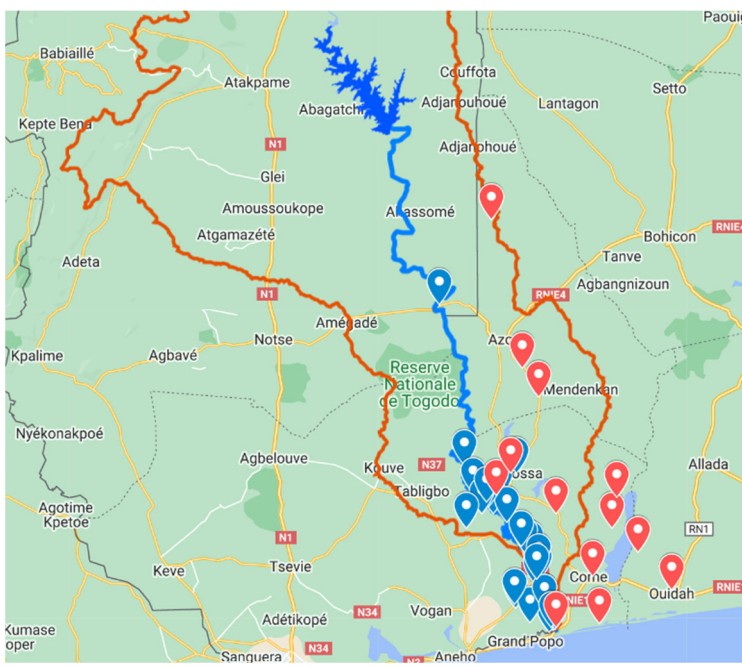

**Figure 4.** Flood hotspots mapping by stakeholders- Benin (red) and Togo (blue).

Part 2. Identification of initial sets of scenarios

The output of this session is to initially identify the most feasible and realistic future scenario that we can simulate with the models. We first presented the climate change scenarios [44,45] and briefly mentioned the task on future land-use scenarios [46]. Lastly, a question was posed on possible other usages of the Adjarala Dam.

First participants were invited to provide feedback on the relevance of the initial set of scenarios. Overall, most of the participants acknowledged that the initial set of scenarios is in line with national climate politics; the answers support as well the integration of the Adjarala Dam in our models and the consideration of years 2030, 2050, and 2100 for flood projections. Participants from Benin suggested additional scenarios for the development of an agriculture project of 400 hectares in Athiémé; construction of Tététou Dam; socio-economic scenario SSP; scenario RCP2.6; dredging of the Ahémé Lake and the Lower Mono. Togo participants suggested the construction of dykes and, dams and reservoirs for agriculture purposes; irrigation of agricultural lands; sand mining in the Mono River; construction of the Tététou Dam; scenario RCP 6.0.

Furthermore, the stakeholders from both countries think that the Adjarala Dam can also be used for the following purposes: tourism, supply of drinking water; fisheries and fish farming; irrigation for agriculture; measurement of discharge and water level; regulation of discharge/flood; water for animals.

Part 3. Identification of initial sets of potential adaptation measures

Here, seven adaptation measures were pre-selected and presented to the stakeholders to guide them through the identification of additional measures. These measures were identified through works of literature and expert judgment based on observation from the 1st Workshop. This first list was amended by the stakeholders based on their local knowledge of the basin and national contexts. Finally, 36 potential adaptation measures were identified by stakeholders from Benin and Togo as the initial set of potential adaptation measures.

Part 4. Identification of initial sets of criteria to evaluate the performance of the measures.

This part was found crucial as it identifies criteria to evaluate the outcome of the measures. Ten criteria were identified in the initial set. To make sure that stakeholders understand perfectly the concept of "criteria" and how to define them, the focus was first put on the first five. It was followed by the ranking of the ten criteria and the identification of new criteria by the stakeholders using Mentimeter.

4.1.3. Step 3-Model Validation and Final Identification of Final Sets of Measures and Criteria

The measures identified in Step 2 (part 3) were synthesized by removing redundant measures and by combining complementary measures. Thus, a final set of eight measures was obtained. In addition, a final list of six criteria was derived based on their relevance and the prioritization work previously conducted by the stakeholders. The final sets of adaptation measures and criteria are presented in Table 2.

**Table 2.** Overview of the final sets of measures and criteria.

| Adaptation Measure | Criteria/Objective |
|---|---|
| 1. Do nothing.<br>2. Retention and detention areas<br>3. Dykes.<br>4. Development of construction and installation plans.<br>5. Development of an early warning system integrating local/natural indicators and establishment of an effective communication system.<br>6. Promotion of flood-resilient, fast-growing crops<br>7. Capacity building of affected communities, researchers, and institutions.<br>8. Insurance. | 1. To reduce the magnitude of surface flooding.<br>2. To minimize the impact of the damage to properties and the agricultural economy.<br>3. To minimize loss of lives.<br>4. To maximize the opportunities for salvaging and recuperation of belongings inside properties and businesses.<br>5. To maximize the acceptance of the public.<br>6. To minimize implementation costs. |

## 4.2. Step 4: Ranking of Measures

This last step, carried out in Workshop 4 through an exercise with stakeholders, demonstrated the consensual decision-making approach for the selection of potential adaptation measures to be implemented. The measures identified in the previous workshop were first presented to pave the way to the main objective of the session. Participants were afterward introduced to the concept of collaborative modeling, the MCDA, and the use of the portal with the decision matrix. Here presented are the results of the groups' weights of criteria, the assessment of the measures, and their subsequent ranking.

### 4.2.1. Criteria Weighting

The weights assigned by stakeholders to criteria are presented in Table 3. Overall, the groups put more weight on the first three criteria than on others. This means that saving lives and sustaining economic activities and income for affected communities are perceived as more important by the stakeholders. In summary, the most to less weighted criteria are minimizing loss of lives, minimizing impacts of damages on properties and agricultural economy, minimizing the magnitude of flood in terms of flood area, and maximizing the acceptance of the public.

**Table 3.** Weight of criteria.

| | Obj 1: To Reduce the Magnitude of Surface Flooding | Obj 2: To Minimize the Impact of the Damage to Properties and Agricultural Economy | Obj 3: To Minimize Loss of Lives | Obj 4: To Maximize the Salvaging and Recuperation of Belongings inside Properties and Businesses | Obj 5: To Maximize the Acceptance of the Public | Obj 6: To Minimize Implementation Cost |
|---|---|---|---|---|---|---|
| Benin | 20 | 25 | 35 | 10 | 5 | 5 |
| Togo | 20 | 20 | 15 | 15 | 20 | 10 |
| Online (Togo-Benin) | 15 | 10 | 30 | 20 | 10 | 15 |

Moreover, stakeholders argued that criterion 6 (To minimize implementation cost) was assigned low weights because cost should not be a main decision element when human lives are at stake.

### 4.2.2. Ranking-Measures vs. Criteria

Measures were assessed based on stakeholders' judgment and perception using a qualitative scale from "very low" to "very high" in a decision matrix. As an example, the decision matrix with the final assessment of Group Benin is presented in Figure 5.

| | ETAPE 2: MATRICE DE DECISION (CLIQUEZ ICI) | | | | | |
|---|---|---|---|---|---|---|
| | **CRITERES / OBJECTIFS** | | | | | |
| | **Obj 1: Réduire l'ampleur des inondations en terme de superficie** | **Obj 2: Minimiser l'impact des dommages sur les propriétés et l'économie agricole** | **Obj 3: Minimiser les pertes en vie humaine** | **Obj 4: Maximiser les possibilités de sauvetage et de récupération des biens à l'intérieur des propriétés et des commerces/entreprises** | **Obj 5: Maximiser l'acceptation du public** | **Obj 6: Minimiser les coûts de mise en œuvre** |
| **MESURES D'ADAPTATION** | km.carré de superficies inondées | Evaluation qualitative: • Très faible • Faible • Moyen • Elevé • Très élevé | Evaluation qualitative: • Très faible • Faible • Moyen • Elevé • Très élevé | Evaluation qualitative: • Très faible • Faible • Moyen • Elevé • Très élevé | Evaluation qualitative: • Très faible • Faible • Moyen • Elevé • Très élevé | Evaluation qualitative: • Très faible • Faible • Moyen • Elevé • Très élevé |
| **Me 1-Ne rien faire** | 910 | Très faible | Très faible | Très faible | Très faible | Très faible |
| **Me 2- Zones de rétention et de détention** | 910 | Très élevé | Elevé | Moyen | Elevé | Faible |
| **Me 3- Digues** | 902 | Très élevé | Elevé | Elevé | Très faible | Moyen |
| **Me 4- Occupation des sols** | 910 | Très élevé | Elevé | Moyen | Faible | Faible |
| **Me 5- Système d'alerte précoce** | 910 | Elevé | Elevé | Elevé | Très élevé | Très faible |
| **Me 6- cultures résistantes aux inondations** | 910 | Elevé | Moyen | Faible | Moyen | Moyen |
| **Me 7- Renforcement des capacités** | 910 | Moyen | Moyen | Moyen | Moyen | Moyen |
| **Me 8- Assurance** | 910 | Moyen | Faible | Moyen | Très faible | Très faible |
| | *Min* | *Min* | *Min* | *Max* | *Max* | *Min* |
| **Poids** | 20 | 25 | 35 | 10 | 5 | 5 |
| EXECUTER TOPSIS | | | | | | |

**Figure 5.** Measures assessment—Group Benin.

The ranking resulting from these assessments based on the TOPSIS method is presented in Table 4 (unadjusted) and Table 5 (adjusted). About Table 4, the analysis of the ranking triggered questions, especially regarding the rank of the "Do nothing" measure at 1st position for Benin and Online groups. Finally, it came out that the criteria were not appropriately framed and induced confusion on the use of scale. For example, "Do nothing" vs. "Minimize loss of lives" was assessed as "very low", meaning that: "There will be very low loss of lives if no adaptation measure is implemented"; however, the groups' members rather meant that: "Doing nothing will have a very low impact on minimizing loss of lives". Thus, they meant that there would be a very high loss of lives if nothing is done, and the appropriate assessment, in that case, should be "very high". Therefore, participants agreed that the assessment needs to be adjusted to account for the new and actual insight (Table 5).

After clarifying the questionable ranking of measures, the assessment of measures vs. criteria was adjusted for some of the criteria that were misunderstood. The first assessments were adjusted inversely with the clarification and consent of the participants. This was done offline by the administrator, and the final adjusted ranking was presented to the stakeholders for further discussion.

Based on the groups' average ranking, the most to less preferred measures are capacity building, dykes, early warning systems, regulation of land use, insurance, and retention zones. Regarding Togo and Benin group ranking, it clearly shows that there are differences in the perceptions and interests of the two participating countries. Togo is leaning more toward soft measures, i.e., capacity building, insurance, and early warning, while Benin prefers hard (physical) measures, i.e., retention and detention zones, effective use of land use, and dykes. The online group that has participants from both Benin and Togo has rankings that

are also different from the country-based groups. This can be attributed to the exchange and dialogue of the different views of the stakeholders from both countries for the assessment of the measures with the criteria. The preference of measures of the online group included both soft and hard measures, i.e., dykes, early warning, and capacity building.

**Table 4.** Unadjusted ranking of measures. The confusion on the description of criteria led to the wrong assessment of measures, subsequently influencing the ranking of measures. Do-nothing was ranked number 1 and 3, generating discussion and needs clarification.

| | Ranking | | |
|---|---|---|---|
| **Measures** | **Benin** | **Togo** | **Online** |
| Me 1-Do nothing | 1 | 3 | 1 |
| Me 2-Retention and detention zones | 7 | 6 | 5 |
| Me 3-Dykes | 6 | 8 | 6 |
| Me 4-Land use planning | 8 | 7 | 3 |
| Me 5-Early warning system | 5 | 5 | 7 |
| Me 6-Flood resilient crops | 4 | 4 | 8 |
| Me 7-Capacity building | 3 | 2 | 4 |
| Me 8-Insurance | 2 | 1 | 2 |

**Table 5.** Adjusted ranking of measures. After clarifying the questionable ranking of measures, the assessment of measures vs. criteria was adjusted for some of the criteria that were misunderstood. The first assessment was adjusted inversely with the clarification and consent of the participants. This was done offline by the administrator, and the final adjusted ranking was presented to the stakeholders for further discussion.

| | Ranking | | |
|---|---|---|---|
| **Measures** | **Benin** | **Togo** | **Online** |
| Me 1-Do nothing | 8 | 8 | 8 |
| Me 2-Retention and detention zones | 1 | 6 | 7 |
| Me 3-Dykes | 3 | 4 | 1 |
| Me 4-Land use planning | 2 | 5 | 6 |
| Me 5-Early warning system | 4 | 3 | 2 |
| Me 6-Flood resilient crops | 5 | 7 | 4 |
| Me 7-Capacity building | 6 | 1 | 3 |
| Me 8-Insurance | 7 | 2 | 5 |

The assessment of the insurance measure was also discussed, and the following points were made by the participants, which were clearly shown through the assessment and ranking of each of the groups:

- Insurance could be accepted by the public if there is good communication/sensitization on it beforehand—Group Togo;
- People will hardly accept insurance options. They might be more open to contributing to a "caisse villageoise" (a local fund established and managed at the village scale) than to subscribe for insurance—Group Benin;
- Insurance should not be among the first options because insurance comes into action only when a disaster strikes. However, the best thing will be to take action to avoid negative impacts and then to minimize the possibilities of needing insurance solutions—Group Online.

In comparison to the Collaborative Modeling framework implemented by Evers et al. (2012) [12] in the non-transboundary catchments in Germany and in the UK, the transboundary LMR case study also successfully increased stakeholders' awareness of flood situations and associated risks and facilitated collaborative discussions on management options. Moreover, the study revealed other similar findings, indicating a lack of clarity regarding the roles of different authorities in FRM and identified instances of miscommunication among them. Therefore, it is essential to improve communication and coordination between local and regional authorities. Despite the favorable results obtained through the proposed approach, obstacles to stakeholder engagement persist, emphasizing the need for the implementation of enduring, customized, and more robust strategies to ensure effective stakeholder involvement. One notable distinction compared to the previous studies is the utilization of various communication tools and strategies to facilitate online-hybrid stakeholder participation. Drawing from our experiences, we have observed the potential of these tools to engage a broader range of stakeholders, particularly in the context of transboundary river basins. Additionally, this approach significantly reduces both the cost and time required to gather stakeholders. However, the challenge lies in maintaining continuous engagement throughout the activity.

## 5. Conclusions

This study considered using the participatory framework Collaborative Modeling for transboundary FRM ranking of measures in the LMR of Togo and Benin. The outcome of the overall process shows that the framework is practicable and adaptable in a transboundary setting leading to consensual decision-making on different aspects of FRM. However, a lengthier engagement process is needed to achieve a more comprehensive and shared understanding of different terms and topics that would lead to more tangible consensual decision-making on the selection of adaptation measures. Findings show that it is important to take note of the phrasing of criteria; it should be more precise to avoid confusion. Our recommendation is to use indicators to describe the assessment, and this may be more appropriate than some criteria descriptions, e.g., "number of deaths" instead of "minimizing loss of lives". The ranking results can also be improved in the future through iterative simulations and more in-depth discussions among all stakeholders. It also shows that Insurance as a flood adaptation measure is perceived differently by stakeholders from the two countries. The online engagement having both Togo and Benin stakeholders can be representative results of transboundary decision-making on the most preferred measures for implementation. This, however, should be taken with caution because of the balance and number of online participants from both countries. Finally, for more consensual decision-making, there is a need for both countries to discuss thoroughly the country-based ranking of results and have another round for all to agree on the criteria weight and the assessment of measures. This has been constrained due to the COVID-19 situation. In this way, social learning will be further enriched to better know and understand the different perceptions to achieve a true consensual decision on the ranking of measures. This would pave the way for more sustainable implementation and maintenance of adaptation measures.

**Author Contributions:** Conceptualization, A.D.S.A. and N.R.H.; methodology, A.D.S.A., N.R.H. and M.E.; data curation, A.D.S.A. and N.R.H.; software, A.D.S.A. and N.R.H.; writing original draft preparation, A.D.S.A. and N.R.H.; writing review and editing, M.E., J.A. and K.K.; funding acquisition, M.E. All authors have read and agreed to the published version of the manuscript.

**Funding:** This research was funded by the German Federal Ministry of Education and Research (BMBF), grant number 01LZ1710B.

**Institutional Review Board Statement:** Not applicable.

**Informed Consent Statement:** Not applicable.

**Data Availability Statement:** Not applicable.

**Acknowledgments:** This research was supported by the CLIENT II-CLIMAFRI project (Implementation of Climate-sensitive Adaptation Strategies to Reduce the Flood Risk in the Catchment Area of the Cross-border Lower Mono River), funded by the German Federal Ministry of Education and Research (BMBF). We are thankful to WASCAL staffs in Benin and Togo for the great support on co-organizing and co-facilitating the stakeholder workshops. We are thankful to the project lead-coordinator United Nations University (UNU) for the outstanding undertaking on organizing the different task and activities leading to the success of the research project.

**Conflicts of Interest:** The authors declare no conflict of interest.

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
