# Peer review of "Transboundary Collaborative Modeling: Consensual Identification and Ranking of Flood Adaptation Measures—A Case Study in the Mono River Basin, Benin, and Togo"

_sustainability, doi:10.3390/su151511728_

Round 1

Reviewer 1 Report

Manuscript number: Sustainability-2445252

Manuscript title: Transboundary Collaborative Modelling: Consensual identification and ranking of flood adaptation measures. A case study 

in the Mono River Basin, Benin and Togo

Dear Authors, I have read and assessed your manuscript.

Well-written manuscript that only needs a few minor revisions to be more complete.

- Figures 2 and 5 require increased spatial resolution

- Line 339, 346, and 360 check typing errors

- Avoid starting sentences with the words "and" and "about" as much as possible

- Tables 4 and 5: excessive use of color. Using both numbers and colors is not advised; the author should utilize numbers to highlight the differences.

- The concluding part shouldn't contain any citations.

Author Response

Reviewer 1:

Manuscript number: Sustainability-2445252

Manuscript title: Transboundary Collaborative Modelling: Consensual identification and ranking of flood adaptation measures. A case study in the Mono River Basin, Benin and Togo

Dear Reviewer,

We would like to express our sincere gratitude for your constructive comments and valuable suggestions on our article. Your expertise and review have greatly contributed to improving the quality and impact of our work. We are pleased to inform you that we have carefully considered and addressed all the suggestions you provided, resulting in a more polished and compelling article. Your insights have helped enrich the content, refine the arguments, and enhance the overall coherence of the piece. Once again, we want to extend my heartfelt appreciation for your time and effort in reviewing my article, and we are confident that your valuable suggestions have significantly strengthened the new version. Please find after this your comments and suggestions that we have addressed.

Sincerely yours

Dr. Adrian Almoradie

Comments and suggestions with our feedback: 

  1. Well-written manuscript that only needs a few minor revisions to be more complete.
  • Thank you for your kind feedback.
  1. Figures 2 and 5 require increased spatial resolution
  • We have replaced the figure with a higher resolution.
  1. Line 339, 346, and 360 check typing errors
  • Space typo corrected
  1. Avoid starting sentences with the words "and" and "about" as much as possible
  • We have made the correction e.g. “And” to “Moreover” in Line 360
  1. Tables 4 and 5: excessive use of color. Using both numbers and colors is not advised; the author should utilize numbers to highlight the differences.
  • We have made the changes without colors
  1. The concluding part shouldn't contain any citations.
  • The citation was deleted.

Reviewer 2 Report

Manuscript Title: Transboundary Collaborative Modelling: Consensual identification and ranking of flood adaptation measures. A case study in the Mono River Basin, Benin and Togo

The research has a very appropriate framework and is well organized.

The following queries have arisen through reviewing the manuscript that need to be considered in the revised version after a MINOR revision.

The initial parts of the abstract are more focused on the topic. Summarize the initial section of the ABSTRACT and present some of main findings.

Provide a classification of collective decision-making approaches and explain which are most appropriate in integrated flood management.

No information has been provided about the land use types/pattern, type of human interventions in the study area (dam construction, flood management measures).

Provide some more information about the amount of precipitation, the type of climate, in introducing of the study region.

Point to the numerical range of the peak flow of floods that occur in the region.

Explain the physiographic features of the area that are effective in intensifying the flood.

The information in Section 3.3 is best presented and organized in a table.

The method of weighting management options seems to be similar to the Delphi approach, in this regard, clarification is needed.

The quality of figure 5 need to be improved.

In the end, mention the limitations of the research.

THE END

Author Response

Reviewer 2:

Manuscript number: Sustainability-2445252

Manuscript title: Transboundary Collaborative Modelling: Consensual identification and ranking of flood adaptation measures. A case study in the Mono River Basin, Benin and Togo

Dear Reviewer,

We would like to express our sincere gratitude for your constructive comments and valuable suggestions on our article. Your expertise and review have greatly contributed to improving the quality and impact of our work. We are pleased to inform you that we have carefully considered and addressed all the suggestions you provided, resulting in a more polished and compelling article. Your insights have helped enrich the content, refine the arguments, and enhance the overall coherence of the piece. Once again, we want to extend my heartfelt appreciation for your time and effort in reviewing my article, and we are confident that your valuable suggestions have significantly strengthened the new version. Please find after this your comments and suggestions that we have addressed.

Sincerely yours

Dr. Adrian Almoradie

Comments and suggestions with our feedback: 

  1. The research has a very appropriate framework and is well organized. The following queries have arisen through reviewing the manuscript that need to be considered in the revised version after a MINOR revision.
  • Thank you for your kind feedback
  1. The initial parts of the abstract are more focused on the topic. Summarize the initial section of the ABSTRACT and present some of main findings.
  • We have substantially revised the abstract. We added more of the findings.
  1. Provide a classification of collective decision-making approaches and explain which are most appropriate in integrated flood management.
  • The definition and its application has been already described in Section 2
  1. No information has been provided about the land use types/pattern, type of human interventions in the study area (dam construction, flood management measures).
  • Information about land-use and human interventions is now provided in section 3.1
  1. Provide some more information about the amount of precipitation, the type of climate, in introducing of the study region.
  • Information about precipitation and climate is now provided in section 3.1
  1. Point to the numerical range of the peak flow of floods that occur in the region.
  • Information about maximum discharge for event 2010 is now provided in section 3.1
  1. Explain the physiographic features of the area that are effective in intensifying the flood.
  • We now describe the flat terrain in the LMR in section 3.1
  1. The information in Section 3.3 is best presented and organized in a table.
  • Thank you for your suggestion. We however think to keep this format to shorten the pages.
  1. The method of weighting management options seems to be similar to the Delphi approach, in this regard, clarification is needed.
  • Thank you for pointing this out. The weighting is not following the Delphi approach; the participant distributes points of a total of 100 based on their own preference. The text in section 3.2 5th paragraph 4th sentence has been further elaborated to make it clear.
  1. The quality of figure 5 need to be improved.
  • We have replaced the figure with a higher resolution.
  1. In the end, mention the limitations of the research.
  • Limitations is described in the discussion and also in the conclusion.

Reviewer 3 Report

Dear Authors,

the paper presents an engaging research which brings some interesting insights to the current state of knowledge and fits to the scope of the journal Sustainability.

What seems particularly interesting to me is the fact that the Collaborative Modelling framework of Evers et. al. was carried out during a Covid-19 pandemic. The pandemic was a big challenge for all procedures requiring the participation of experts or the public.

While I think the article is good, I took the liberty of suggesting a few changes that may further improve the quality of the work.

1.     Please add some additional details about the Mono river (hydrological regime, minimum, average, maximum flows etc.) and the Mono river basin (land use structure, runoff, population, biggest cities etc.).

2.     Please explain what language was used during the workshops and indicate what are the official languages in the countries analysed.

3.     Please expand the thread of online works and describe in more detail. What problems have you encountered, e.g. technical? How long did online meetings last compared to live meetings? What should be secured, what should you pay attention to during online workshops? Who ran the workshops? What was his education and professional profile?

4.     Please indicate what made the decision when choosing online technical tools. Is it possible to use other similar applications? Which one for example?

5.     Please improve the quality of the figures. 5. It is illegible.

6.     I suggest expanding the discussion by referring to other river basins where the Evers at al. method has been applied.

Best regards.

Author Response

Reviewer 3:

Manuscript number: Sustainability-2445252

Manuscript title: Transboundary Collaborative Modelling: Consensual identification and ranking of flood adaptation measures. A case study in the Mono River Basin, Benin and Togo

Dear Reviewer,

We would like to express our sincere gratitude for your constructive comments and valuable suggestions on our article. Your expertise and review have greatly contributed to improving the quality and impact of our work. We are pleased to inform you that we have carefully considered and addressed all the suggestions you provided, resulting in a more polished and compelling article. Your insights have helped enrich the content, refine the arguments, and enhance the overall coherence of the piece. Once again, we want to extend my heartfelt appreciation for your time and effort in reviewing my article, and we are confident that your valuable suggestions have significantly strengthened the new version. Please find after this your comments and suggestions that we have addressed.

Sincerely yours

Dr. Adrian Almoradie

Comments and suggestions with our feedback: 

  1. While I think the article is good, I took the liberty of suggesting a few changes that may further improve the quality of the work.
  • Thank you for your kind feedback
  1. Please add some additional details about the Mono river (hydrological regime, minimum, average, maximum flows etc.) and the Mono river basin (land use structure, runoff, population, biggest cities etc.).
  • We have now described some of the important geographical, hydrological and climate information in section 3.1
  1. Please explain what language was used during the workshops and indicate what are the official languages in the countries analysed.
  • The language used in the workshop was French which is officially spoken by both countries. This is now mentioned in section 3.2 1st paragraph
  1. Please expand the thread of online works and describe in more detail. What problems have you encountered, e.g. technical? How long did online meetings last compared to live meetings? What should be secured, what should you pay attention to during online workshops? Who ran the workshops?What was his education and professional profile?
  • We have now described this in section 3.2 4th paragraph
  1. Please indicate what made the decision when choosing online technical tools. Is it possible to use other similar applications?Which one for example?
  • We have now described this in section 3.2 4th paragraph
  1. Please improve the quality of the figures. 5. It is illegible.
  • We have replaced the figure with a higher resolution.
  1. I suggest expanding the discussion by referring to other river basins where the Evers at al. method has been applied. .
  • We have provided a paragraph with the insights in section 4 last paragraph.

Reviewer 4 Report

I have read with interest your paper on Transboundary Collaborative Modelling: Consensual identification and ranking of flood adaptation measures. A case study in the Mono River Basin, Benin and Togo. Manuscript is characterized by clear structure and right proportions of the components.

Authors report the results of transboundary cooperation related to flood risk management in general. It is based on the study case of the Mono river basin.

The literature review as well as conclusion and discussion section should be improved and developed.

There is no sufficient data for comparability with other authors’ results.

In my opinion, presented manuscript might be published as a technical report, not as an article.

Author Response

Reviewer 4:

Manuscript number: Sustainability-2445252

Manuscript title: Transboundary Collaborative Modelling: Consensual identification and ranking of flood adaptation measures. A case study in the Mono River Basin, Benin and Togo

Dear Reviewer,

We would like to express our sincere gratitude for your comments and suggestions on our article. Please find after this your comments and suggestions that we have addressed.

Sincerely yours

Dr. Adrian Almoradie

Comments and suggestions with our feedback: 

  1. The literature review as well as conclusion and discussion section should be improved and developed.
  • We have revised the section following the suggestion of other reviewers
  1. There is no sufficient data for comparability with other authors’ results.
  • From our perspective, the results, experiences, and insights obtained from our study offer a valuable comparison to the two non-transboundary case studies conducted by Evers et al. (2012). We have now included these insights in Section 4. There we highlighted the significance of our findings comparing to the research of the other case studies.
  1. In my opinion, presented manuscript might be published as a technical report, not as an article.
  • We strongly believe that this article is significant in the field of science because it underscores the importance of integrating physical, social, and technological factors in the spheres of planning and decision-making for transboundary basins. Insufficient research has been conducted in this area when it comes to engaging stakeholders. By bridging these domains, we aim to provide a comprehensive perspective that contributes to the advancement of knowledge. We genuinely hope that this revised version will positively influence your perception of our contribution. To improve the article, we have carefully considered the input from the other reviewers and incorporated additional information into the revised manuscript.

Round 2

Reviewer 4 Report

I accept manuscript in present form, however, I still think that it is rather technical report than an article

Author Response

Dear Reviewer,

Thank you for reviewing the manuscript. 

Kind regards

Dr. Adrian Almoradie